# Salivary Metabolites in Breast Cancer and Fibroadenomas: Focus on Menopausal Status and BMI

**DOI:** 10.3390/metabo14100531

**Published:** 2024-09-30

**Authors:** Elena I. Dyachenko, Lyudmila V. Bel’skaya

**Affiliations:** Biochemistry Research Laboratory, Omsk State Pedagogical University, 644099 Omsk, Russia; dyachenko.ea@gkpc.buzoo.ru

**Keywords:** breast cancer, fibroadenomas, leaf-shaped (phyllodes) tumor, saliva, biochemistry, menopausal status, BMI, diagnosis

## Abstract

This study of the features of the biochemical composition of biological fluids in patients with breast cancer, including saliva, allows us to identify some indicators as metabolic predictors of the presence of the disease. Objectives: to study the influence of the menopause factor and body mass index (BMI) on the biochemical composition of saliva and to evaluate the applicability of metabolic markers of saliva for the diagnosis of breast cancer. Methods: The case–control study involved 1438 people (breast cancer, *n* = 543; fibroadenomas, *n* = 597; control, *n* = 298). A comprehensive study of the biochemical composition of saliva was carried out using 36 parameters. Results: When comparing the salivary biochemical composition in breast cancer, fibroadenomas, and controls, it is necessary to take into account the menopausal status, as well as BMI (less than 25 or more) for the group of patients with preserved menstrual function. A complex of biochemical parameters has been identified that change in saliva during breast cancer, regardless of menopause and BMI (total protein, urea, uric acid, NO, α-amino acids, GGT), as well as specific parameters that must be taken into account when analyzing individual subgroups (imidazole compounds, LDH, catalase, α-amylase). During the study of a separate group of patients with leaf-shaped (phyllodes) tumors, we found similarities with breast cancer in the changes in some biochemical parameters that can be attributed to metabolites of malignant growth (protein, α-amino acids, calcium, NO, pyruvate, peroxidase, α-amylase). Conclusions: We demonstrated changes in a wide range of salivary biochemical parameters depending on the presence of fibroadenomas and breast cancer. From the point of view of clinical practice, this may be useful information for monitoring the condition of patients with fibroadenomas, which are difficult to unambiguously classify based on instrumental diagnostics alone.

## 1. Introduction

Breast pathologies can be classified according to a wide range of etiologies with malignant and benign diagnoses [1]. Invasive ductal carcinoma is the most common malignant lesion [2,3], while fibroadenomas are the most common solid benign breast lesions, with a prevalence of 15–23% [4]. Breast cancer (BC) comprises a group of diseases characterized by various biological subtypes, molecular profiles, and specific clinicopathological characteristics [5]. Comprehensive BC assessment includes not only anatomical data, but also important prognostic data, such as the expression of estrogen receptors (ERs) and progesterone receptors (PRs), human epidermal growth factor receptor 2 (HER2), and available gene expression data [6,7,8,9]. Fibroadenomas (FAs) are characterized by the proliferation of epithelial and stromal components, usually clearly defined and distinct from the surrounding breast tissue [10]. These benign neoplasms are a mixture of glandular and connective components. There is also a leaf-shaped tumor (leaf-shaped fibroadenoma, phyllodes tumor), which occupies an intermediate position between fibroadenomas and breast cancer and can progress to cancer in 3–5% of cases [11,12].

Current diagnostic methods often have difficulty differentiating cancer from fibroadenoma [13]. To differentiate fibroadenomas, phyllodes tumors, and breast cancer, artificial intelligence and machine learning technologies can be used in biopsy differentiation [14], mammography, and ultrasound image processing [10,15], as well as genetic methods [16,17] and in combinations of different methods [18,19,20].

The study of the biochemical composition of biological fluids in patients with breast cancer allows us to identify certain indicators as metabolic predictors of the presence of the disease [21,22]. The study of metabolic manifestations of the systemic effect of a malignant tumor on the body allows us to consider changes in enzyme activity, the content of substrates, and products of enzymatic reactions as “metabolic markers” of malignant growth [23]. Breast cancer and fibroadenomas are two independent diseases with different histological and molecular genetic characteristics [24]. An analysis of the metabolic processes in breast cancer and fibroadenomas allows us to highlight the difference between the two pathological conditions and describe the reasons for including complex forms of fibroadenomas in the list of risk factors for breast cancer.

An analysis of metabolic changes in breast cancer can be performed using a number of biological fluids, including blood plasma [21,22], urine [25,26,27], etc. Currently, saliva is considered as an informative biological fluid that allows for tracking changes in the metabolic markers of oncological diseases [28,29,30]. The advantages of saliva are non-invasive collection, no risk of infection when working with biomaterial, and high information content [31]. However, biochemical indicators, even in practically healthy people, significantly depend on factors such as age, menopause status, and body mass index (BMI) [32,33,34]. In this study, we examined the influence of menopause and BMI on the biochemical composition of saliva and assessed the applicability of saliva metabolic markers for diagnosing breast cancer.

## 2. Materials and Methods

### 2.1. Study Design

The case–control study involved 1438 people, including the main group (breast cancer, *n* = 543), comparison group (fibroadenomas, *n* = 597) and control group (healthy control, *n* = 298). Patients of the main group and comparison group were hospitalized for planned surgical treatment in the absence of biopsy or for the first course of chemotherapy in the presence of biopsy results. After histological verification, patients were assigned to the appropriate group. None had received any prior treatment, including hormone therapy, chemotherapy, molecularly targeted therapy, radiotherapy, surgery, etc. The inclusion criteria were considered as follows: a patient 30–70 years of age, the absence of any treatment at the time of the study, the absence of signs of active infection (including purulent processes), and good oral hygiene. The volunteers included in the study did not reveal any clinically significant concomitant diseases other than cancer pathology (in particular, diabetes mellitus, cardiovascular pathologies, etc.) that could affect the results of the study. The exclusion criteria were a lack of histological verification of the diagnosis.

All patients were Caucasian. The structure of the study groups is shown in Table 1.

All patients in the main group had histologically and cytologically confirmed invasive breast carcinoma of the following stages: stage I—135 (24.7%), stage II—239 (44.0%), stage III—112 (20.6%), stage IV—59 (10.7%). In 303 patients, no signs of metastasis to regional lymph nodes were found (N_0_—55.8%); in 240 patients, metastases were found in the displaced axillary lymph nodes (N_1–3—_44.2%). TNM classification was carried out in accordance with AJCC (8th edition, 2017). Breast tumors were classified by the degree of tissue differentiation into highly and moderately differentiated (GI + II, *n* = 102), and poorly differentiated (GIII, *n* = 268). In all cases, the status of HER2, estrogen, and progesterone receptors was determined. In 394 patients (72.6%), HER2-negative status was confirmed, in 94 (17.3%)—HER2-positive; in 156 patients (28.7%), ER-negative status was confirmed, in 335 (61.7%)—ER-positive; in 225 patients (41.4%), PR-negative status was confirmed, in 265 (48.8%)—PR-positive. Ki-67 values less than 20% were determined in 173 patients (31.9%), more than 20%—in 317 patients (58.4%). According to the molecular biological subtypes of breast cancer, the patients were distributed as follows: triple negative (TNBC)—97 (17.8%), luminal A—123 (22.7%), luminal B (HER2-negative)—174 (32.0%), luminal B (HER2-positive)—47 (8.7%), non-luminal BC—47 (8.7%). No mammary gland pathologies were detected in the volunteers of the control group during routine mammographic and ultrasound examinations.

### 2.2. Collection, Processing, Storage and Analysis of Saliva Samples

Saliva (5 mL) was collected from all participants prior to treatment. The collection of saliva samples was carried out on an empty stomach after rinsing the mouth with water in the interval of 8–10 am by spitting into sterile polypropylene tubes; the salivation rate (mL/min) was calculated. We did not find significant differences in the salivary flow rate in the studied groups. Saliva samples were centrifuged (10,000× *g* for 10 min) (CLb-16, Moscow, Russia), after which biochemical analysis was immediately performed without storage and freezing using the StatFax 3300 semi-automatic biochemical analyzer (Awareness Technology, Palm City, FL, USA). Between receipt at the laboratory and the testing, saliva samples were placed in a refrigerator at 2–8 °C (for no more than 24 h).

A total of 36 biochemical parameters were determined in all saliva samples. The pH, mineral composition (calcium, phosphorus, sodium, potassium, magnesium, chlorides), the content of urea, total protein, albumin, uric acid, α-amino acids, imidazole compounds, seromucoids, sialic acids, and the activity of enzymes (aminotransferases, alkaline phosphatase, lactate dehydrogenase, gamma-glutamyl transferase, α-amylase) were determined in all samples. The content of substrates for lipid peroxidation processes (diene conjugates, triene conjugates, Schiff bases, malondialdehyde) was determined. Additionally, we assessed the activity of antioxidant enzymes (catalase, superoxide dismutase, antioxidant activity, and peroxidase).

The pH of human saliva was measured in triplicate for each participant. The pH meter was accurate to ±0.002 pH units and a three-point calibration was used at pHs 4, 7, and 10. The concentration of potassium, sodium and magnesium ions (mmol/L) was determined using the KAPEL-105M capillary electrophoresis system (Lumex, St. Petersburg, Russia). The total calcium content (mmol/L) was determined photometrically by reaction with Arsenazo III, phosphorus (mmol/L) by the reaction of molybdenum acid ammonium, and chlorides (mmol/L) by reaction with mercury thiocyanate using the kits from Vektor-Best LLC (Novosibirsk, Russia). Urea concentration (mmol/L) was determined photometrically by the Bertlot urease-salicylate method, total protein (g/L) by reaction with pyrogallol red, albumin (g/L) by reaction with green bromocresol, uric acid (UA, nmol/mL) by uricase method, and sialic acids (SAs, mmol/L) according to the Hess method. The total content of α-amino acids (α-AAs, mmol/L) was determined by reaction with ninhydrin; imidazole compounds (ICs, mmol/L) were determined by orange-red coloration with a diazo reagent in an alkaline medium. Alkaline phosphatase activity (ALP, U/L) was determined by the Bessey–Lowry–Brock endpoint method, lactate dehydrogenase (LDH, U/L) by the kinetic ultraviolet method according to the NADH (Nicotinamide Adenine Dinucleotide) oxidation rate, gamma glutamyl transferase (GGT, U/L) by the kinetic method using L-gamma-glutamyl-3-carboxy-4-nitroanilide as a Zeitz–Persin substrate, and α-amylase (U/L) by the kinetic method for the hydrolysis of CNP-oligosaccharide with the formation of 2-chloro-4-nitrophenol. The determination of aminotransferase activity (ALT, AST, U/L) was carried out using the unified Reitman–Frenkel method; the de Ritis ratio was additionally calculated (AST/ALT, c.u.). The activity of superoxide dismutase (SOD, c.u.) was determined by the accumulation of the product of auto-oxidation of adrenaline by the superoxide anion radical in an alkaline medium, and catalase (CAT, mcat/L) by the rate of decrease in hydrogen peroxide in the incubation medium. Peroxidase activity was determined by the blue coloration during the oxidation reaction of benzidine in the presence of hydrogen peroxide (c.u.). Antioxidant activity (АОА, mmol/L) was determined by recording the rate of oxidation of the reduced form of 2,6-dichlorophenolindophenol by oxygen dissolved in the reaction medium. The content of substrates for lipid peroxidation processes (diene conjugates—DC, triene conjugates—TC, Schiff bases—SB, c.u.) was determined spectrophotometrically by the Volchegorsky method. The content of malondialdehyde (MDA, nmol/mL) was determined by a reaction with thiobarbituric acid. The level of middle molecular toxins was determined by ultraviolet spectrophotometry at wavelengths of 254 and 280 nm, and the ratio MM 280/254 nm was calculated (MM, c.u.). The level of seromucoids (SM, c.u.) was determined using the turbidimetric method, according to Huergo. Pyruvic acid (PYR, mmol/L) condensed with 2,4-dinitrophenylhydrazine to form a hydrazone, which in an alkaline medium gave a brown-red solution with subsequent spectrophotometric determination. To determine lactate (mmol/L), a method was used based on the appearance of a yellow color when adding lactic acid to iron (III) sulfate in a 0.02% sulfuric acid solution. The nitric oxide (NO, μmol/L) was determined by a photometric method using the Griess reagent.

### 2.3. Statistical Analysis

The distribution and homogeneity of variances in the groups were preliminarily checked. According to the Shapiro–Wilk test, the content of all the parameters being determined does not correspond to the normal distribution (*p* < 0.05). The conducted test for the homogeneity of variances in the groups (Bartlett’s test) allowed us to reject the hypothesis that the variances are homogeneous across the groups (*p* < 0.0001). Therefore, nonparametric statistical methods were used to process the obtained data.

The values in the tables are given as a relative change in the parameter compared to the corresponding control group. Relative changes are calculated as the difference between the corresponding value for breast cancer/fibroadenomas and healthy controls relative to healthy controls (%).

Statistical analysis was performed using Statistica 13.3 EN software (StatSoft, Tulsa, OK, USA); R version 3.6.3; RStudio Version 1.2.5033; FactoMineR version 2.3. (RStudio, version 3.2.3, Boston, MA, USA) by a nonparametric method using the Mann–Whitney U-test and the Kruskal–Wallis H-test. The description of the sample was made by calculating the median (Me) and the interquartile range as the 25th and 75th percentiles [LQ; UQ]. Differences were considered statistically significant at *p* ˂ 0.05.

A principal component analysis (PCA) was performed using the PCA program in R [35]. PCA results are presented in the form of factor planes and corresponding correlation circles. In each case, the figures show only the first two principal components (PC1 and PC2). The color of the arrows on the correlation circle changes from blue (weak correlation) to red (strong correlation) as shown on the color bar. The orientation of the arrows characterizes positive and negative correlations (for the first principal component, we analyze the location of the arrows relative to the vertical axis; for the second principal component, relative to the horizontal axis). The significance of the correlation is determined by the correlation coefficient (r): strong—r = ±0.700 to ±1.00, medium—r = ±0.300 to ±0.699, weak—r = 0.00 to ±0.299.

## 3. Results

### 3.1. The Influence of Age and Menopausal Status on the Biochemical Composition of Saliva

The age structure of the study groups is extremely heterogeneous. Thus, fibroadenomas predominate in the age groups of 20–29, 30–39 and 40–49 years, while the proportion of patients with breast cancer increases after 50 years (Figure 1A). At the first stage, we tested the effect of age on the value of salivary biochemical parameters. For this purpose, we identified subgroups by age in 10-year increments (20–29, 30–39, 40–49, 50–59, 60–69 and 70+) on the combined group (*n* = 1438). The PCA method showed that the 20–29, 30–39 and 40–49 year old groups were located on the factor diagram to the left of the vertical axis “0-0”, while the 50–59, 60–69 and 70+ groups were located to the right (Figure 1B). The subgroups aged 50–59 and 60–69 were the closest to each other. The subgroup aged 70+ was represented on the diagram by the largest ellipse, since it is the smallest and the data have a large spread (Figure 1B).

We assumed that patients could be grouped by age into two groups: 20–49 years and 50+, taking into account the average age of menopause (Figure 1B,C). The PCA method showed that the subgroups “20–49 years” and “No menopause”, as well as “50+” and “Menopause” did not coincide in the factor diagram (Figure 1D). Thus, the presence/absence of menopause shifted the centers of gravity of the ellipses in the diagram (Figure 1C,D). This indicates that it is not the age of the patients that should be taken into account, but the menopause status (Figure 1D). When only the menopause status was taken into account, the division in the factor diagram became more complete (Figure 1C).

Next, we identified a list of biochemical parameters that change in saliva regardless of the presence of menopause (Appendix A). These parameters include total protein, urea, uric acid, total α-amino acid content, NO, and GGT activity (Table 2). The relative change in concentration depends on the pathology: breast cancer or fibroadenoma. Thus, the concentrations of protein, urea, and uric acid decrease to a greater extent in breast cancer. The total α-amino acid content and GGT activity increase more strongly in breast cancer, while the concentration of NO increases more strongly in fibroadenomas (Table 2).

Some parameters changed in breast cancer and fibroadenomas only before or after menopause. For example, the concentration of electrolyte components (calcium, chlorides), the activity of AST, ALP, LDH, α-amylase and peroxidase, and lipid peroxidation indices (DC, SB, MDA) changed in saliva in breast cancer and fibroadenomas before menopause. After menopause, the number of significant biochemical parameters decreased; in particular, the activity of ALT, catalase and AOA, as well as the content of sialic acids, changed significantly (Table 2). Statistically significant changes regarding sialic acids were manifested only in the postmenopausal period and were multidirectional. Thus, in the group with breast cancer, we observed an increase in concentration, whereas in fibroadenomas, there was a decrease in the content of sialic acids (Table 2).

When comparing pairwise subgroups with BC, fibroadenomas, and healthy controls before/after menopause, it was shown that in each pair there were statistically significant differences in the concentrations of albumin (*p* = 0.0486, *p* < 0.0001 and *p* = 0.0001 for BC, fibroadenomas, and healthy controls, respectively), imidazole compounds (*p* = 0.0005, *p* = 0.0001 and *p* = 0.0221), and catalase activity (*p* = 0.0263, *p* = 0.0159 and *p* = 0.0009), as well as pH (*p* = 0.0007, *p* = 0.0053 and *p* = 0.0001), which allows us to associate these biochemical parameters with menopause status. Although protein concentration (*p* = 0.0163, *p* = 0.0053 and *p* = 0.0013 for BC, fibroadenomas, and healthy control, respectively) had statistically significant differences, significant changes were observed depending on the presence and nature of the pathological disease. For other indicators, single changes were observed depending on the presence of menopause, which may be due to the sample characteristics (Appendix A).

### 3.2. Comparison of Subgroups with Breast Cancer, Fibroadenomas, and Healthy Controls, Taking into Account the Presence/Absence of Menopause

It was shown that before menopause, the biochemical composition of saliva in breast cancer and fibroadenomas had similar values, but significantly differed from the healthy control (*p* < 0.0001) (Figure 2A). After menopause, significant differences were observed in the subgroup with breast cancer from the healthy control and fibroadenomas (*p* < 0.0001) (Figure 2B).

The contribution of individual biochemical parameters to the division of subgroups in the factor diagram before and after menopause was different (Figure 2C,D). Thus, before menopause, in contrast to post menopause, the contribution of LDH (*r* = 0.4816) and antioxidant protection parameters, including catalase (*r* = 0.4714), AOA (*r* = 0.6663) and peroxidase (*r* = 0.6046), was significant (Table 3). After menopause, the significant parameters included pH (*r* = 0.5426), uric acid (*r* = 0.4583), ALP (*r* = −0.4387), and GGT (*r* = −0.4571).

### 3.3. The Influence of BMI on the Biochemical Composition of Saliva

At the next stage, the contribution of BMI to the values of biochemical parameters of saliva was assessed (Figure 3A). It was shown that the subgroup with normal BMI (<25) differed significantly, while no differences were found between the subgroups of BMI = 25–30 and BMI > 30. To understand which factor caused more differences between the subgroups (menopause or BMI), we conducted a comparison, taking into account the influence of both factors (Figure 3B). It was found that for the subgroup without menopause, the influence of BMI was significant, while after menopause, the influence of this factor was leveled.

It was shown that some biochemical parameters changed significantly when menopause status and BMI were taken into account simultaneously (Appendix A). Moreover, more changes were noted for the fibroadenomas subgroup (Table 4). For both BC and fibroadenomas, the factors dependent on BMI were pH, calcium concentration, urea, albumin, imidazole compounds and GGT activity (Table 4). For BC, BMI significantly affected α-amylase activity: for the subgroup with BMI < 25, α-amylase activity was 7 times higher than in the other subgroups. With BMI < 25, lipid peroxidation processes were less pronounced, the level of medium molecular toxins was lower, the concentration of sodium, potassium, and chlorides was lower, but peroxidase activity was higher (Table 4). A decrease in lactate concentration and an increase in pyruvate concentration were observed in the subgroup with BMI < 25, which is why the lactate/pyruvate ratio decreased for this subgroup.

Thus, when comparing the biochemical composition of saliva in breast cancer, fibroadenomas, and healthy controls, it is necessary to take into account menopausal status, as well as BMI (less than 25 or more) for the group of patients with preserved menstrual function. 

### 3.4. Biochemical Composition of Saliva in Phyllodes Fibroadenomas

We identified a subgroup of phyllodes tumors, which included 55 patients, 50 of whom had not reached menopause at the time of examination, so the comparison was made with the subgroups of fibroadenomas, BC, and healthy controls before menopause (Table 5, Appendix A).

Phyllodes tumors were closer to BC in terms of total protein, total α-amino acid content, calcium, NO, pyruvate, peroxidase and α-amylase activity (Table 5). For chlorides, uric acid, imidazole compounds, alkaline phosphatase activity, GGT and SOD, phyllodes tumors were closer to fibroadenomas. Phyllodes tumors were unique in some respects: sodium concentration was higher; and magnesium and potassium were lower than in the other groups. The concentration of urea and sialic acids decreased significantly, but the level of lactate and diene conjugates increased (Table 5). According to a number of indicators, phyllodes tumors were located between fibroadenomas and BC (Table 5).

## 4. Discussion

There is currently no doubt that differential diagnostics of breast cancer and fibroadenomas is extremely important in the clinical context [19,36]. Therefore, it is very important to find simple and quick indicators that will help doctors differentiate between benign and malignant breast tumors. The biochemical indicators of saliva can also be included in such indicators. 

We have shown that the total protein level decreased in the group of patients with fibroadenomas and breast cancer compared to the control group, while with the presence of menopause and high BMI, we observed a relative increase in its concentration in saliva. Against the background of a general decrease in the concentration of proteins in saliva, a relative increase in the concentration of albumin and α-AAs was observed. The lowest albumin concentration was observed in the group of patients with breast cancer in the postmenopausal period, with a slight decrease in fibroadenomas, while in the premenopausal period its highest concentration was in the group of patients with breast cancer. We saw a direct relationship between BMI and albumin concentration only in the premenopausal period. In turn, the content of α-AAs directly correlated with the presence of postmenopausal status in patients with both fibroadenomas and breast cancer, and with BMI in the group of patients with fibroadenomas. Moreover, the highest concentrations were found in the breast cancer group. A decrease in the albumin content in saliva may be due to a combination of factors such as the presence of an inflammatory process and insufficiency of the amino acid composition used for albumin synthesis. We assume a redistribution of free amino acids, which may be associated with ensuring the vital activity of cancer cells. The proposed assumption is consistent with data on the maximum decrease in albumin in breast cancer, an increase in the concentration of α-AAs and the activity of aminotransferases, which are involved in the redistribution of nitrogen in the body [37].

High urea content was noted among patients with pathology compared to the control group. The maximum value was in the group of patients with breast cancer. Urea concentration had a direct correlation with the increase in BMI in both breast cancer and fibroadenomas. The concentration of uric acid with the presence of menopause was sharply reduced among patients with breast cancer and correlated with an increase in BMI and the presence of the menopausal period. Among patients with fibroadenomas, the maximum concentration of uric acid occurred in the premenopausal period. We noted that in the postmenopausal period, any changes in the concentration of uric acid were leveled by age-related and hormonal changes. This may be due to the effect of estrogens on renal clearance and excretion of urates in the urine [38].

The change in the concentration of urea towards its increase reflects the predominance of catabolic processes over anabolic ones, which also correlates with a decrease in the content of protein and albumin [39]. The presence of a high concentration of uric acid reflects a high consumption of purine bases to maintain the proliferative activity of cells [40]. It is interesting that the highest content of uric acid is in the group of patients with fibroadenomas. There may be several reasons for this: the activation of the immune system, and the active proliferation of epithelial-connective tissue cells of fibroadenoma. Thus, with regard to protein metabolism, we note different biochemical processes underlying the vital activity of cells in fibroadenomas and breast cancer. Based on the data obtained, we can assume that proliferative processes prevail more in fibroadenomas, while in breast cancer we see a predominance of destructive processes.

Changes in salivary aminotransferase activity before menopause directly depended on the BMI value in the premenopausal period in both the group of patients with breast cancer and fibroadenomas, i.e., the activity increased. In the postmenopausal period, the activity values of all aminotransferases were significantly lower than the values taking into account the high BMI. The maximum and statistically significant increase in concentration was observed in the group of patients with breast cancer. At the same time, in fibroadenomas, we also noted a reliable increase in aminotransferase activity compared to the control. Since aminotransferases are non-specific markers, their activity can change in various pathological processes. In this case, we assume that the main contribution to the increase in aminotransferase activity occurs due to the redistribution of nitrogen in the body, and the active transport of amino acids to target zones, i.e., for the life support of cancer cells or for the proliferation of cells in fibroadenomas [41,42]. At the same time, we do not exclude the influence of possible concomitant pathology.

Imidazole compounds are the structural core of the essential amino acid histidine, which is a product of histamine degradation, an inducer of immunological processes [43], and part of the purine bases of adenine and guanine [44]. In our study, we observed a statistically significant decrease in the level of imidazole compounds in groups, breast cancer and fibroadenoma, with the most pronounced decrease in fibroadenomas. According to the data obtained, the content of imidazole compounds in saliva has a reliable dependence on BMI.

The enzyme α-amylase is primarily known for its participation in the breakdown of carbohydrates [45]. Recently, a sufficient number of studies have accumulated that reflect the multifunctionality of the enzyme. Thus, α-amylase can act as a marker of stressful conditions in the body on a par with the hormone cortisol [46] as a result of the activation of the sympathoadrenergic system [47,48]. A recent experimental study showed that it is salivary α-amylase that can have an antiproliferative effect on cells in breast cancer [49]. The salivary gland and mammary gland have a common embryology, and just like the salivary gland, they produce α-amylase [50]. Mammary gland cells are sensitive to the effects of salivary α-amylase. One explanation for the effect of α-amylase on cell proliferation is interference with hormones that stimulate growth, which include estrogens. Increased α-amylase levels had an inhibitory effect on estradiol binding to its receptors [51]. This is consistent with our data, where α-amylase had the highest concentration in breast cancer, especially among premenopausal patients. Phyllodes tumors had an intermediate concentration of α-amylase in saliva between fibroadenomas and breast cancer.

In fibroadenomas and breast cancer, we recorded a reliable change in the following agents involved in oxidative processes: an increase in NO, peroxidase, DC, CAT, MDA and a decrease in SB, AOA. It is noteworthy that in the group of patients with fibroadenomas, the maximum increase in NO, MDA and a decrease in AOA was shown compared to the breast cancer group. It is known that NO is a highly reactive free radical in biological systems, capable of combining with other free radicals, molecular oxygen and heavy metals [52,53]. In addition to damaging the phospholipid layer of the cell membrane, NO can form peroxynitrite (ONOO-) and N_2_O_3_, which can lead to damage to the DNA structure due to its biological ability to oxidize and nitrate DNA, as well as causing a break in single-stranded DNA by attacking the sugar–phosphate backbone. NO levels have been associated with the activation or inhibition of cancer genesis in a number of studies [54]. Peroxidase activity increases the formation of free radical compounds that damage the cellular structure [55]. The stages of lipid peroxidation can be judged by the accumulation of DC (refers to the first link of lipid oxidation) [56], SB (the second link of lipid oxidation) [57], and MDA (the final product of lipid oxidation) [58]. AOA and CAT reflect the reserve capacity of the biological system to suppress cascade oxidative reactions [59], which are depleted with age. According to the data obtained, we can assume that in fibroadenomas, lipid peroxidation occurs more intensively compared to the breast cancer group, as evidenced by more noticeable changes in the levels of NO, MDA, and SB, which are accompanied by a more pronounced depletion of antioxidant protection.

SM, also known as orosomucoid (ORM1) or alpha-1-acid glycoprotein (AGP), belongs to the group of acute phase proteins, has anti-inflammatory and immunomodulatory effects by suppressing the activity of neutrophils and components of the complement system, and has an inhibitory effect on IL-1 through the activation of macrophages [60,61]. According to literary sources, SM can actively participate in angiogenesis by the autocrine pathway through SM synthesized by endothelial cells, and by the paracrine pathway through circulating forms both in typical pathological processes and in oncological diseases, in particular in breast cancer [62,63]. Terminal sialic acid residues (SAs) on the SM surface are key components in the activation of reactive oxygen species in neutrophils stimulated by the chemotactic peptide fMLP (N-Formylmethionine-leucyl-phenylalanine) and affect the mobilization of Ca^2+^ in their cytoplasm. It was also shown that the desialylated form of SM loses its activity in stimulating reactive oxygen species in neutrophils [64]. A significant change towards an increase in SM is observed in the group of patients with fibroadenomas and pre-menopausal breast cancer, with the maximum increase in the group with fibroadenomas. We assume that in fibroadenomas there is a more pronounced activation of reactive oxygen species production and stimulation of the immune response than in breast cancer. Thus, in breast cancer, the SM level is lower than in fibroadenomas, but the SA content is higher, which indicates destructive processes occurring in mucoproteins and the presence of less reactive forms of SM than in fibroadenomas.

Electrolyte imbalances are observed in many pathological processes, including cancer. Most often, changes affect the levels of potassium, sodium, magnesium, and calcium in almost all biological fluids [65]. The causes of imbalance in cancer patients include paraneoplastic syndrome, impaired diuresis, tumor lysis syndrome, and concomitant clinical conditions accompanied by inflammatory processes [66]. Electrolyte imbalances can trigger serious pathology associated with dysfunction of many organs [67]. It has been shown that K^+^ outflow is important for the sustained influx of Ca^2+^ into immune cells through Ca^2+^-permeable channels. It maintains membrane potential depolarization, maintaining an electrochemical gradient for further Ca^2+^ influx [68]. We observed the lowest free calcium values in phyllodes tumors and breast cancer, against the background of high extracellular potassium content in saliva. In this case, this may reflect the active consumption of free calcium by cells in exchange for potassium, which can affect both the calcification and destruction of cellular structures with the release of potassium into the extracellular space. It is known that patients with leaf-shaped tumors have a higher risk of developing breast cancer [69]. Leaf-shaped tumors are characterized by increased tissue calcification [70]. Low levels of free calcium can also reflect the activity of inflammatory processes. The trigger for tissue calcification can be lipid oxidation, reactive oxygen species, inflammatory cytokines, etc. [71]. It is known that for the activation of neutrophils, the activation of reactive oxygen species in neutrophils and their degranulation, and the activation of β2-integrins, adhesive capacity, and migration are necessary [72]. Also, the increase in SM in fibroadenomas and their effect on the regulation of the activation of the immune response were discussed above. Thus, it has been shown that SM can induce the mobilization of Ca^2+^ in the cytoplasm of neutrophils [73].

Electrolyte concentrations (calcium and potassium), pH, albumin, urea, imidazole compounds, and GGT statistically significantly depended on BMI. The listed markers are not specific to any particular pathological condition. In our study, we showed that high BMI makes a significant contribution to the deviation in these markers regardless of the pathological condition, which means that the patient’s BMI should be taken into account when analyzing changes in the level of biochemical indicators.

Our study demonstrated the difference in biochemical processes occurring in cells of fibroadenomas, phyllodes tumors and breast cancer. Thus, the characteristic metabolic parameters for fibroadenomas include increased levels of NO, uric acid, and MDA, and decreased levels of α-AAs, GGT, ALP, amylase, DC, lactate, peroxidase. It is interesting that in some biochemical parameters, phyllodes tumors are close to either breast cancer (α-AAs, NO) or fibroadenomas (GGT, ALP). Noticeable differences in the metabolism of phyllodes tumors are demonstrated by a decrease in the level of chlorides, CAT, and urea, and an increase in the content of DC and lactate. In terms of the level of protein, UA, MDA, α-amylase, and peroxidase, phyllodes tumors have an intermediate concentration between breast cancer and fibroadenomas. The data we obtained allowed us to more thoroughly trace the difference between malignant and benign formations, and take a new look at general biochemical indicators, which in some cases can be regarded as markers of aggressive processes occurring in the body with breast cancer.

The limitations of the study include consideration of the influence of only biochemical parameters on metabolic processes in breast cancer and fibroadenomas, while it is interesting to trace whether the identified patterns extend to hormonal status, the level of tumor markers, cytokines, amino acids, etc. In this article, we also did not consider the influence of the clinical, pathological and molecular biological characteristics of breast cancer on the values of biochemical parameters of saliva, which will be carried out at the next stages of the study.

## 5. Conclusions

In this study, we demonstrated changes in a wide range of salivary biochemical parameters depending on the presence of benign or malignant neoplasms, such as fibroadenomas and breast cancer. We found that for fibroadenomas, metabolic processes reflect active inflammatory processes. At the same time, in breast cancer, we observed a less “acute” immune response and more pronounced proliferative processes due to high concentrations of α-amylase and amino acid redistribution.

Reliable data have been obtained showing that a number of salivary biochemical markers directly depend on BMI and menopausal status. Therefore, when analyzing a patient’s biochemical result, it is necessary to take into account the contribution of these factors to changes in some biochemical parameters (imidazole compounds, LDH, catalase, α-amylase, etc.). A number of salivary biochemical parameters change in breast cancer regardless of the presence of menopause and BMI, namely total protein, urea, uric acid, NO, total content of α-amino acids, and GGT.

In a study of a separate group of patients with phyllodes tumors, we found similarities with breast cancer in changes in some biochemical parameters that can be attributed to metabolites of malignant growth (total protein content, total content of α-amino acids, calcium, NO, pyruvate, peroxidase and α-amylase activity). From the point of view of clinical practice, this may be useful information for monitoring the condition of patients with fibroadenomas, which are difficult to unambiguously classify based on instrumental diagnostics alone.

## Figures and Tables

**Figure 1 metabolites-14-00531-f001:**
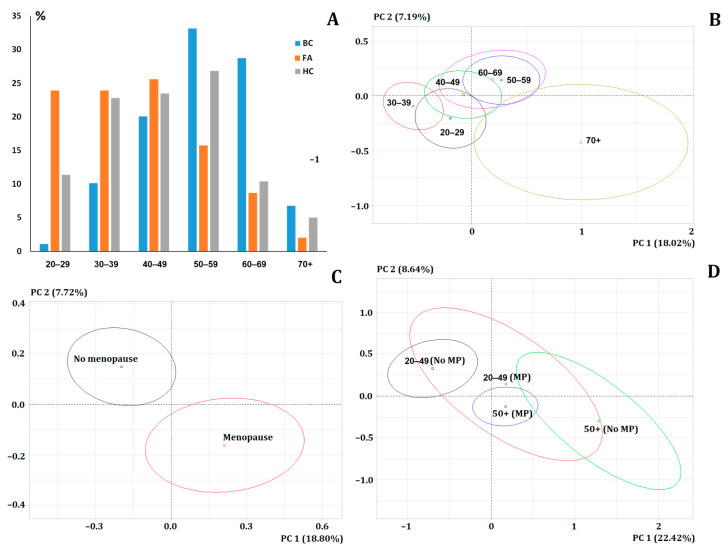
The effect of age and menopause status on the biochemical composition of saliva: (**A**) Age structure of the study groups (%). BC—breast cancer, FA—fibroadenomas, HC—healthy control. (**B**) PCA depending on the age group (*p* < 0.0001). (**C**) PCA depending on the menopause status (*p* = 0.0002). (**D**) PCA depending on the presence/absence of menopause in the age groups “20–49 years” and “50+” (*p* = 0.0012). MP—menopause. The loadings of the PCA plot for (**B**–**D**) are given in Appendix A.

**Figure 2 metabolites-14-00531-f002:**
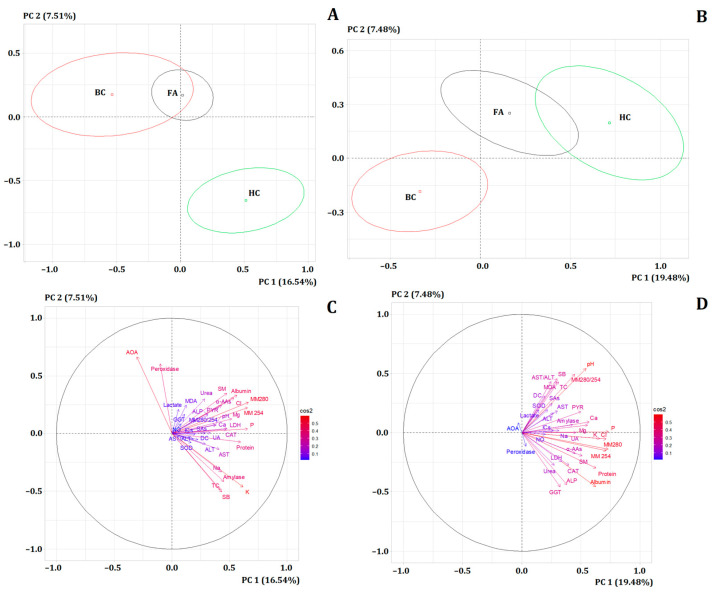
PCA of three subgroups: breast cancer, fibroadenomas and healthy controls: (**A**) Before menopause. (**B**) After menopause. (**C**) The contribution of individual biochemical parameters to the separation of groups before menopause. (**D**) The contribution of individual biochemical parameters to the separation of groups after menopause. BC—breast cancer, FA—fibroadenomas, HC—healthy controls.

**Figure 3 metabolites-14-00531-f003:**
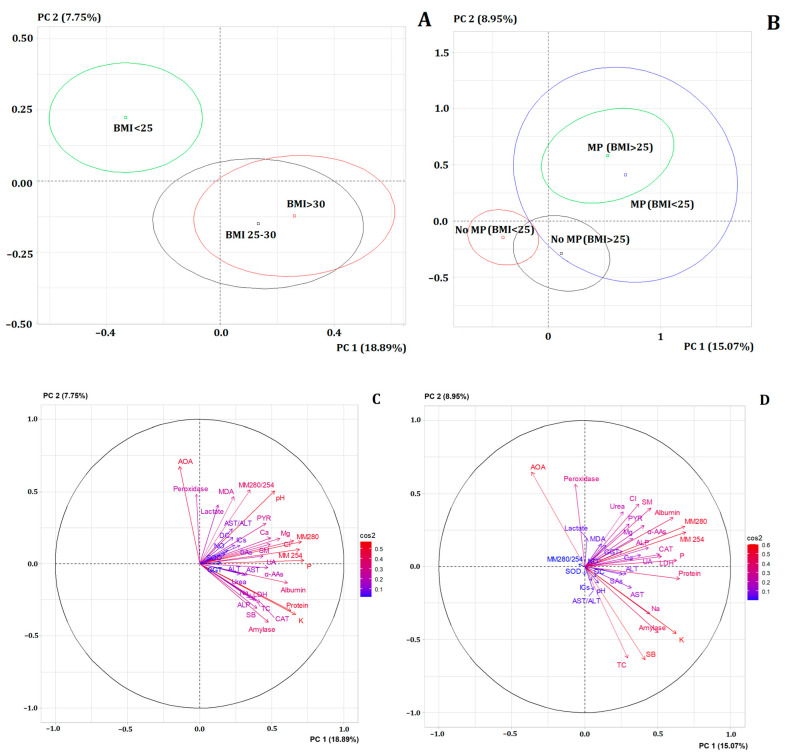
(**A**) РСА depending on BMI, (**B**) РСА depending on BMI and menopausal status. МР—menopause, (**C**) the loadings of the PCA plot for the BMI subgroup analysis, (**D**) the loadings of the PCA plot for the BMI menopausal status subgroup analysis.

**Table 1 metabolites-14-00531-t001:** Structure of study groups by age, menopause status and BMI.

Subgroups	Breast Cancer, *n* = 543	Fibroadenomas, *n* = 597	Healthy Control, *n* = 298
Age			
20–29 years	6 (1.1%)	143 (24.0%)	34 (11.4%)
30–39 years	55 (10.1%)	143 (24.0%)	68 (22.8%)
40–49 years	109 (20.0%)	153 (25.6%)	70 (23.5%)
50–59 years	181 (33.3%)	94 (15.7%)	80 (26.8%)
60–69 years	156 (28.7%)	52 (8.7%)	31 (10.4%)
70+ years	37 (6.8%)	12 (2.0%)	15 (5.1%)
Menopause			
Yes	382 (70.3%)	168 (28.1%)	142 (47.7%)
No	161 (29.7%)	429 (71.9%)	156 (52.3%)
Body Mass Index (BMI)			
<25	135 (24.7%)	291 (48.7%)	201 (67.4%)
25–30	179 (33.0%)	148 (24.8%)	97 (32.6%)
>30	229 (42.3%)	158 (26.5%)	-

**Table 2 metabolites-14-00531-t002:** Relative change in the biochemical composition of saliva depending on menopause status for breast cancer and fibroadenomas compared with the corresponding healthy control, %.

Indicator	No Menopause		Kruskal–Wallis Test (H, p)	Menopause		Kruskal–Wallis Test (H, p)
	BC	FA		BC	FA	
рН	−0.2	0.8	4.450; 0.1081	−0.9	−0.4	1.796; 0.4074
Са	−6.8	−0.9	6.160; 0.0460 *	−0.7	1.9	1.667; 0.5579
P	6.2	2.4	0.8417; 0.6565	2.3	−5.4	5.001; 0.0821
Na	−19.6	−14.5	3.513; 0.1727	−5.4	−20.9	3.419; 0.1810
K	−7.1	−6.3	4.570; 0.1018	12.4	7.7	0.6792; 0.7121
Cl	4.8	−3.9	7.717; 0.0211 *	−1.7	−4.8	1.894; 0.3879
Mg	−1.3	−3.6	4.288; 0.1172	−0.4	−5.4	1.950; 0.3772
Protein	−43.4	−35.0	49.13; 0.0000 *	−43.8	−39.9	55.47; 0.0000 *
Urea	28.5	15.5	17.35; 0.0002 *	52.4	38.9	49.79; 0.0000 *
UA	−14.6	16.4	7.395; 0.0248 *	−37.5	−24.2	18.27; 0.0001 *
Albumin	23.4	14.6	2.643; 0.2668	−8.6	4.2	1.581; 0.4537
ALT	1.9	0.0	1.789; 0.4089	8.2	−1.0	6.977; 0.0306 *
AST	12.5	9.4	6.225; 0.0445 *	5.6	1.4	4.737; 0.0936
AST/ALT	6.6	9.2	3.533; 0.1709	0.7	−5.3	1.630; 0.4425
α-AAs	3.1	2.0	11.85; 0.0027 *	3.2	2.5	7.051; 0.0294 *
ICs	−1.1	−6.8	0.7383; 0.6913	−5.4	−8.1	3.344; 0.1878
NO	25.6	45.0	27.10; 0.0000 *	24.4	41.9	9.591; 0.0083 *
ALP	25.9	18.5	15.18; 0.0005 *	17.2	15.5	3.352; 0.1871
MM 254	−9.1	−12.7	3.367; 0.1857	−8.4	−8.2	2.276; 0.2624
MM 280	−10.6	−6.0	1.411; 0.4939	−12.9	−14.2	3.435; 0.1795
LDH	40.8	37.9	12.64; 0.0018 *	23.2	12.8	4.148; 0.1257
CAT	−17.0	−15.4	3.442; 0.1789	−20.2	−18.7	14.18; 0.0008 *
SAs	3.3	0.0	3.536; 0.1707	13.3	−6.7	7.295; 0.0261 *
PYR	16.3	4.1	3.263; 0.1956	5.3	0.0	0.5329; 0.7661
DC	2.0	0.3	8.514; 0.0142 *	0.9	1.8	0.2152; 0.8980
TC	−2.5	1.1	4.502; 0.1053	0.9	−1.0	3.115; 0.2106
SB	−2.0	1.8	8.736; 0.0127 *	−0.5	−0.4	0.0755; 0.9630
MDA	5.3	9.3	14.85; 0.0006 *	3.8	9.0	3.301; 0.1919
GGT	18.5	11.5	24.90; 0.0000 *	9.2	0.9	12.82; 0.0016 *
SM	13.3	21.1	10.37; 0.0056 *	−4.9	2.9	0.5916; 0.7439
SOD	18.2	4.5	1.843; 0.3980	21.7	8.7	4.442; 0.1085
α-Amylase	101.1	40.6	13.09; 0.0014 *	29.8	3.8	2.003; 0.3673
Lactate	−8.4	−9.5	4.520; 0.1043	6.4	5.8	0.0558; 0.9725
AOA	−11.4	−7.4	3.522; 0.1719	−3.6	−13.0	11.72; 0.0028 *
Peroxidase	104.1	69.4	7.538; 0.0231 *	−22.2	−9.3	0.7821; 0.6764
MM280/254	2.0	4.7	4.565; 0.1020	0.8	1.4	0.5087; 0.7754

Note. BC—breast cancer, FA—fibroadenomas, Ca—calcium, P—phosphorus, Na—sodium, K—potassium, Cl—chlorides, Mg—magnesium, UA—uric acid, ALT—alanine aminotransferase, AST—aspartate aminotransferase, α-AAs—total α-amino acid content, ICs—Imidazole compounds, NO—nitric oxide, ALP—alkaline phosphatase, MM—medium molecular weight toxins, LDH—lactate dehydrogenase, CAT—catalase, SAs—sialic acids, PYR—pyruvic acid, DC—diene conjugates, TC—triene conjugates, SB—Schiff bases, MDA—malondialdehyde, GGT—gamma glutamyl transferase, SM—seromucoids, SOD—superoxide dismutase, AOA—antioxidant activity. *—differences between the 3 groups are statistically significant (BC, FA and HC). Blue color shows the increase in concentration of the biochemical indicator, red color shows the decrease in concentration of the corresponding parameter. The more intense the color, the more significant the changes.

**Table 3 metabolites-14-00531-t003:** Correlation coefficients of individual biochemical parameters for separating three groups (breast cancer, fibroadenomas and healthy controls).

Indicators	No Menopause		Indicators	Menopause	
PC 1 (First principal component)					
MM 280	0.6638 *	6.006 × 10^−96^	P	0.7344	2.037 × 10^−118^
MM 254	0.6577	1.219 × 10^−93^	MM 280	0.7278	2.684 × 10^−115^
P	0.6575	1.473 × 10^−93^	MM 254	0.7127	1.660 × 10^−108^
K	0.6180	8.982 × 10^−80^	Cl	0.7120	3.212 × 10^−108^
Protein	0.5952	1.104 × 10^−72^	K	0.6584	2.537 × 10^−87^
Albumin	0.5644	5.497 × 10^−64^	Protein	0.6276	3.651 × 10^−77^
Cl	0.5383	2.872 × 10^−57^	Albumin	0.6242	4.075 × 10^−76^
Mg	0.5204	5.281 × 10^−53^	Са	0.5700	5.748 × 10^−61^
LDH	0.4816	1.382 × 10^−44^	Mg	0.5568	1.142 × 10^−57^
SM	0.4758	2.037 × 10^−43^	рН	0.5426	2.606 × 10^−54^
CAT	0.4714	1.528 × 10^−42^	SM	0.5095	4.789 × 10^−47^
α-Amylase	0.4496	2.160 × 10^−38^	PYR	0.4999	4.285 × 10^−45^
SB	0.4301	6.086 × 10^−35^	UA	0.4583	2.734 × 10^−37^
α-AAs	0.4297	7.199 × 10^−35^	α-AAs	0.4526	2.659 × 10^−36^
Na	0.4296	7.505 × 10^−35^	MM280/254	0.4462	3.265 × 10^−35^
TC	0.4197	3.414 × 10^−33^	α-Amylase	0.4289	2.183 × 10^−32^
AST	0.4114	7.790 × 10^−32^			
PYR	0.4067	4.351 × 10^−31^			
PC 2 (Second principal component)					
AOA	0.6663	6.535 × 10^−97^	рН	0.5432	1.935 × 10^−54^
Peroxidase	0.6046	1.559 × 10^−75^	MM280/254	0.4923	1.377 × 10^−43^
α-Amylase	−0.4146	2.399 × 10^−32^	SB	0.4568	5.012 × 10^−37^
K	−0.4634	5.544 × 10^−41^	MDA	0.4316	8.251 × 10^−33^
TC	−0.4858	1.942 × 10^−45^	AST/ALT	0.4312	9.385 × 10^−33^
SB	−0.5046	1.945 × 10^−49^	TC	0.4294	1.832 × 10^−32^
			ALP	−0.4387	5.711 × 10^−34^
			Albumin	−0.4547	1.152 × 10^−36^
			GGT	−0.4571	4.484 × 10^−37^

Note. *—Only strong and medium strength correlation coefficients are given.

**Table 4 metabolites-14-00531-t004:** Relative change in the biochemical parameters of saliva in breast cancer and fibroadenomas depending on menopause and BMI compared to the corresponding control, %.

Indicator	Breast Cancer			Kruskal–Wallis Test (H, p)	Fibroadenomas			Kruskal–Wallis Test (H, p)
	No Menopause		MP		No Menopause		MP	
	BMI < 25	BMI > 25			BMI < 25	BMI > 25		
рН	0.0	−0.6	−0.9	10.97; 0.0042 *	1.3	0.2	−0.4	10.36; 0.0056 *
Са	−9.6	−6.8	−0.7	9.605; 0.0082 *	−2.1	2.8	1.9	8.164; 0.0169 *
P	0.1	7.4	2.3	3.905; 0.1420	2.2	2.4	−5.4	1.254; 0.5342
Na	−33.1	0.3	−5.4	4.438; 0.1087	−21.9	−0.2	−20.9	5.237; 0.0729 **
K	−13.6	−4.0	12.4	8.819; 0.0122 *	−8.5	−1.5	7.7	5.278; 0.0714
Cl	−4.9	12.2	−1.7	4.562; 0.1022	−6.3	−1.6	−4.8	16.64; 0.0002 *^,^**
Mg	−1.8	−1.5	−0.4	2.067; 0.3558	−2.8	−5.5	−5.4	3.155; 0.2065
Protein	−43.3	−44.5	−43.8	5.752; 0.0564	−36.6	−32.1	−39.9	13.29; 0.0013 *^,^**
Urea	16.7	51.8	52.4	20.94; 0.0000 *^,^**	8.7	23.3	38.9	20.93; 0.0000 *
UA	−13.2	−18.6	−37.5	3.0494 0.2177	19.3	10.8	−24.2	2.303; 0.3162
Albumin	14.0	31.1	−8.6	6.307; 0.0427 *	7.1	20.7	4.2	22.20; 0.0000 *^,^**
ALT	0.0	5.8	8.2	0.5789; 0.7489	−1.9	3.8	−1.0	1.601; 0.4492
AST	11.7	21.9	5.6	0.0098; 0.9951	7.8	13.3	1.4	0.2695; 0.8795
AST/ALT	16.8	1.0	0.7	1.173; 0.5563	11.8	4.4	−5.3	3.060; 0.2165
α-AAs	2.0	4.5	3.2	2.642; 0.2668	1.6	3.3	2.5	10.95; 0.0042 *
ICs	2.3	−2.3	−5.4	13.80; 0.0010 *	−5.7	−9.1	−8.1	16.87; 0.0002 *
NO	25.2	16.8	24.4	0.1113; 0.9458	48.5	38.2	41.9	0.7927; 0.6728
ALP	22.2	37.0	17.2	0.7731; 0.6794	14.8	25.9	15.5	3.991; 0.1360 **
MM 254	−19.0	−7.2	−8.4	3.297; 0.1923	−18.3	−5.5	−8.2	8.005; 0.0183 *^,^**
MM 280	−13.0	−2.9	−12.9	3.274; 0.1945	−8.9	−1.7	−14.2	6.167; 0.0454 *
LDH	40.8	43.4	23.2	2.348; 0.3091	27.1	45.7	12.8	2.690; 0.2605
CAT	−10.6	−20.4	−20.2	5.373; 0.0681	−17.0	−10.6	−18.7	6.046; 0.0487 *
SAs	6.7	3.3	13.3	0.3909; 0.8225	−3.3	6.7	−6.7	1.562; 0.4580
PYR	24.5	14.3	5.3	1.621; 0.4447	4.1	8.2	0.0	3.849; 0.1459
DC	2.6	1.1	0.9	5.051; 0.0800	0.2	0.7	1.8	0.0599; 0.9705
TC	−2.5	−3.3	0.9	6.548; 0.0378 *	0.1	2.6	−1.0	5.249; 0.0725
SB	−1.3	−4.0	−0.5	2.376; 0.3049	2.1	1.8	−0.4	1.918; 0.3832
MDA	8.0	2.7	3.8	1.212; 0.5455	9.3	10.0	9.0	0.0327; 0.9838
GGT	13.1	20.4	9.2	6.673; 0.0356 *^,^**	11.0	12.6	0.9	6.994; 0.0303 *
SM	25.3	3.6	−4.9	2.605; 0.2719	20.5	21.7	2.9	0.8607; 0.6503
SOD	15.9	22.7	21.7	0.6256; 0.7314	4.5	2.3	8.7	1.228; 0.5411
α-Amylase	204.6	28.8	29.8	4.351; 0.1136 **	45.3	21.0	3.8	0.6253; 0.7315
Lactate	−23.4	−2.0	6.4	4.606; 0.0999	−11.1	−9.1	5.8	3.323; 0.1898
AOA	−8.7	−14.7	−3.6	6.286; 0.0432 *	−6.2	−10.2	−13.0	5.764; 0.0560
Peroxidase	128.6	75.5	−22.2	1.876; 0.3914	79.6	38.8	−9.3	3.979; 0.1367
MM280/254	2.0	2.3	0.8	0.0994; 0.6067	5.6	3.0	1.4	1.674; 0.4330

Note. *—the differences between the three groups are statistically significant (“BMI < 25 + No MP”, “BMI > 25 + No MP”, “MP”), **—differences between the groups “BMI < 25 + No MP”, “BMI > 25 + No MP” are statistically significant, *p* < 0.05. МР—menopause. Blue color shows the increase in concentration of the biochemical indicator, red color shows the decrease in concentration of the corresponding parameter. The more intense the color, the more significant the changes.

**Table 5 metabolites-14-00531-t005:** Relative change in the biochemical parameters of saliva in fibroadenomas, phyllodes tumors and breast cancer compared to healthy controls, %.

Indicator	Fibroadenomas	Phyllodes Tumors	Breast Cancer	Kruskal–Wallis Test (H, p)
рН	0.6	0.8	−0.2	3.740; 0.2909
Са	−0.1	−4.6	−6.8	5.866; 0.1183
P	2.7	1.9	6.2	1.305; 0.7280
Na	−16.2	−0.7	−19.6	6.040; 0.1097
K	−5.0	−11.0	−7.1	5.458; 0.1412
Cl	−3.6	−6.3	4.8	13.07; 0.0045 *
Mg	−2.8	−10.7	−1.3	6.741; 0.0806
Protein	−34.6	−44.2	−43.4	33.24; 0.0000 *
Urea	20.4	−4.6	28.5	24.86; 0.0000 *
UA	16.4	15.1	−14.6	12.72; 0.0053 *
Albumin	15.4	5.2	23.4	1.307; 0.7275
ALT	0.0	−1.9	1.9	5.360; 0.1473
AST	9.4	25.0	12.5	5.942; 0.1145
AST/ALT	9.2	10.2	6.6	0.8219; 0.8442
α-AAs	2.0	3.7	3.1	16.97; 0.0007 *
ICs	−6.8	−9.1	−1.1	2.859; 0.4139
NO	45.0	27.5	25.6	32.75; 0.0000 *
ALP	18.5	18.5	25.9	13.16; 0.0043 *
MM 254	−12.7	−6.8	−9.1	1.773; 0.6208
MM 280	−5.8	−6.8	−10.6	3.078; 0.3797
LDH	40.1	14.6	40.8	3.962; 0.2656
CAT	−16.7	−10.9	−17.0	28.02; 0.0000 *
SAs	0.0	−3.3	3.3	2.719; 0.4370
PYR	2.0	23.5	16.3	5.865; 0.1184
DC	0.0	4.2	2.0	12.39; 0.0062 *,**
TC	1.1	0.1	−2.5	7.290; 0.0632
SB	1.8	1.8	−2.0	6.078; 0.1079
MDA	9.3	7.3	5.3	17.02; 0.0007 *
GGT	11.5	12.6	18.5	9.987; 0.0187 *
SM	20.5	24.1	13.3	5.095; 0.1650
SOD	4.5	4.5	18.2	2.105; 0.5509
α-Amylase	37.1	68.2	101.1	11.77; 0.0082 *
Lactate	−11.9	3.2	−8.4	7.789; 0.0506 *,**
AOA	−9.7	1.4	−11.4	5.299; 0.1511
Peroxidase	63.3	91.8	104.1	9.692; 0.0214 *
MM280/254	4.7	5.3	2.0	3.308; 0.3465

Note. *—differences between four groups are statistically significant (fibroadenomas, phyllodes tumors, breast cancer, healthy control), **—differences between fibroadenomas and phyllodes tumors are statistically significant. Blue color shows the increase in concentration of the biochemical indicator, red color shows the decrease in concentration of the corresponding parameter. The more intense the color, the more significant the changes.

## Data Availability

The raw data supporting the conclusions of this article will be made available by the authors on request.

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
