# Peer review of "Salivary Metabolites in Breast Cancer and Fibroadenomas: Focus on Menopausal Status and BMI"

_metabolites, 2024, doi:10.3390/metabo14100531_

Round 1

Reviewer 1 Report

Comments and Suggestions for Authors

     In this manuscript, the authors report studies aimed at exploring the influence of menopause and body mass index (BMI) on the biochemical composition of saliva in breast cancer patients and to evaluate whether changes in metabolic markers in saliva could have applicability in the diagnosis of breast cancer.  The authors report that comparisons of the biochemical composition in saliva need to take into account menopause status and BMI.  The authors report on a set of metabolites that change in breast cancer, fibroadenomas and phyllodes tumors. They suggest that these changes may be useful in helping monitor the conditions of patients with fibromatosis.

            This is an interesting study looking at changes in metabolites in saliva in patients with breast cancer and fibroadenomas and the effect of menopausal status and BMI.  Although the manuscript is well written, the amount of data (36 metabolites) and the number of groups (cancer, fibroadenomas, phylloides, menopause or not, normal or high BMI), make it not very straightforward to follow. Overall, the study makes a positive contribution to the field.  There are several concerns and suggestions for improvement:

1.      The manuscript does not provide the patient and control characteristics from all three groups, in terms of demographics, menopause status, BMI and other characteristics.  A Table summarizing this information should be included.

2.      The methodology section needs to be a little more specific regarding the assays and methodology/kits used for the substances assayed.  Presently, it just provides the name of the instrument. 

3.      The manuscript indicates that saliva samples were “immediately” assayed without storage and freezing.  What was the interval between obtaining sample collection and processing?  Were samples at least maintained at 4oC before and during assay? 

4.      Tables in the main manuscript provide only the relative changes as % compared to controls.  It was not possible for the reviewer to access the Supplementary material with the actual values for the metabolites in all groups with median and interquartile ranges.  Were the % calculated on the control group median values?

5.      The authors conclude that metabolic processes in fibroadenomas reflect, and active inflammatory process compared to breast cancer.  Determination of the concentrations of pro-inflammatory cytokines would have helped support this conclusion.   

6.      One of the main conclusions is that monitoring metabolite changes may be helpful in following up patients with fibroadenomas. A small table summarizing which metabolites, and the direction of change would be helpful to finalize the discussion and support the conclusion.

Reviewer 2 Report

Comments and Suggestions for Authors

The study of the biochemical composition of biological fluids in patients with breast cancer allows us to identify certain indicators as metabolic predictors of the presence of the disease.

Provide references.

In the introduction, please mention the use of other biofluids for tracking metabolic indicators in oncological studies.

What were the loadings of the PCA plot? The authors should perform an OPLS-DA plot for the PCA plot corresponding to 1C because there is a fair amount of overlap between the groups.

The protein content seems to be altered in cancer. The authors can quantitate protein in the urine to see the clearance status.

Authors must perform an OPLS-DA plot for Figure 3A-B.

Round 2

Reviewer 1 Report

Comments and Suggestions for Authors

In this version, the authors have adequately addressed the critiques of the reviewer.